# Detection of Household Furniture Storage Space in Depth Images

**DOI:** 10.3390/s22186774

**Published:** 2022-09-07

**Authors:** Mateja Hržica, Petra Pejić, Ivana Hartmann Tolić, Robert Cupec

**Affiliations:** Faculty of Electrical Engineering, Computer Science and Information Technology Osijek, 31000 Osijek, Croatia

**Keywords:** storage space detection, neural network, depth images

## Abstract

Autonomous service robots assisting in homes and institutions should be able to store and retrieve items in household furniture. This paper presents a neural network-based computer vision method for detection of storage space within storage furniture. The method consists of automatic storage volume detection and annotation within 3D models of furniture, and automatic generation of a large number of depth images of storage furniture with assigned bounding boxes representing the storage space above the furniture shelves. These scenes are used for the training of a neural network. The proposed method enables storage space detection in depth images acquired by a real 3D camera. Depth images with annotations of storage space bounding boxes are also a contribution of this paper and are available for further research. The proposed approach represents a novel research topic, and the results show that it is possible to facilitate a network originally developed for object detection to detect empty or cluttered storage volumes.

## 1. Introduction

Service robots operating in unstructured indoor environments such as homes, hospitals, and schools should be able to independently perform various object manipulation tasks. These tasks often include placing objects inside furniture, e.g., putting utensils in cabinets, putting books on a bookshelf, or placing office supplies on a desk. To perform such tasks, the robot must be able to recognize the spaces inside or on the top of furniture that are suitable for storing items. In this paper, we propose an approach which uses a visual sensor and artificial intelligence for detection of cuboidal storage space in storage furniture.

### 1.1. The Problem and the Contributions

In this paper, a storage space above a horizontal surface that can hold various items is referred to as a *storage volume*. We consider detection of cuboid storage volumes, which can be represented by bounding boxes. The image processing method presented in this paper is able to detect storage volumes in a single depth image of household furniture. For a practical application, successful detection from a single depth image is of great importance because it is impractical for a robot in a realistic scenario to obtain many images from different views and reconstruct the whole model in order to detect storage space. Furthermore, in a realistic environment, storage furniture is rarely empty. It usually holds one or more items, which are easily added, removed, and often change position. Therefore, it is important to recognize storage volumes regardless of their current occupancy status. In this paper, we propose an approach for detection of storage volumes in terms of the storage capacity of a particular piece of furniture, not in terms of the amount of empty storage space currently available.

The most successful methods for object detection are based on machine learning, and more recently on neural networks (NN). This implies training of an artificial neural network and transferring its knowledge to a mobile robot agent with a visual sensor. Training a neural network requires a large dataset with properly labeled regions of interest that the robot should recognize. Since the aim of this study was to recognize household 3D storage volumes, the training dataset had to consist of 3D scenes with household storage furniture with annotated storage volumes. To our knowledge, none of the available 3D indoor scenes datasets have such annotations. Therefore, to achieve our goal, we needed to create these annotations on an appropriate dataset. Manual annotation of the ground truth is prone to human error and time consuming; thus, we propose a method for automated annotation of 3D storage volumes in synthetic depth images.

The proposed procedure whose overview is given in Figure 1, consisting of two novel methods, detects storage volumes in depth images even if these volumes are cluttered by items. The first method starts from empty 3D mesh models of household furniture, such as bookshelves, cabinets, and tables, and automatically provides ground truth annotations of the bounding boxes of their storage volumes. Such models can be found in a number of publicly available 3D model databases, such as ShapeNet [1]. The 3D models with assigned ground truth bounding boxes representing storage volumes are used as input to the second method. This method generates realistic depth images of furniture with objects, such as boxes, bottles, books and lamps, representing realistic clutter commonly found in households and randomly placed on the shelves. This approach allows fast and fully automatic generation of a large number of synthetic depth images with information about the bounding boxes of storage volumes which can be used as the training dataset for neural networks. To demonstrate the feasibility of our approach, we trained the VoteNet [2] and FCAF3D [3] neural networks, originally developed for object recognition, to detect storage volumes. The trained network was finally evaluated on a test dataset consisting of synthetic and real depth images with assigned ground truth bounding boxes of storage volumes. The images acquired by a real 3D camera were manually annotated using annotation software which we developed for the purpose of the presented research. It should be noted that these manually annotated images were used only for testing purposes, and the neural network training was performed using exclusively automatically generated synthetic data.

The contributions of this paper are:An empty storage volume detection algorithm which provides automatic ground truth annotation of storage volumes in empty synthetic 3D furniture mesh models.A method for generating depth images of realistic synthetic scenes containing storage furniture with randomly cluttered storage space.Experimental evaluation of detection of storage volumes inside or on the top of furniture in depth images using the VoteNet and FCAF3D neural networks where the storage volumes could contain items.A dataset containing synthetic and real depth images with annotated ground truth storage volume bounding boxes.

### 1.2. Paper Overview

The paper is structured as follows. Section 2 comprehends the state of the art methods related to the object detection in depth images, and more precisely, detection of empty volumes in storage furniture. In Section 3, two methods are presented, which represent the contributions of this paper: (I) a method for automatic detection of storage volume bounding boxes in 3D models of furniture, and (II) a method for generating realistic synthetic depth images of scenes containing furniture models with automatically annotated storage volume bounding boxes for the purpose of machine learning. Section 4 describes training of the VoteNet and FCAF3D neural networks and evaluation of the trained network on both synthetic and real depth images. Furthermore, in that section, acquisition of real depth images and their ground truth labeling are described. The results of the evaluation of the proposed approach are presented in Section 5. Detailed overview of the results and comparison of the storage volume detector’s performance on synthetic and real datasets are presented in Section 6. The paper is finally concluded in Section 7.

## 2. Related Research

The research presented in this paper focuses on the detection of empty and cluttered storage space above furniture shelves in depth images. We investigated solutions to this problem by applying methods commonly used for object detection—more specifically, neural network-based 3D object detectors. In recent literature, researchers have proposed solutions for object detection in 2D and 3D visual sensor data [4,5,6]. We focus on obtaining 3D information to facilitate robotic manipulation. Robots used for autonomous manipulation typically rely on depth-view sensors such as RGB-D cameras and LiDARs to provide point-cloud representations of 3D scenes. Newer 3D object detectors that use point clouds are typically based on deep learning methods that require large datasets with annotated objects of interest for training [2,3,7,8,9,10].

In our work, we investigated the application of neural network detectors for storage volume detection in household furniture. We considered VoteNet [2] and FCAF3D [3]. VoteNet, similarly to H3DNet [7], uses a voting scheme for generating object proposals. This approach is applied to many object recognition problems. For example, 3D object recognition in point clouds using the Hough voting model is proposed in [5]. VoteNet is used to vote on the centers of objects and learn to aggregate the votes according to their features and local geometry to generate high quality object proposals. The importance of multi-level context information for 3D object recognition from point cloud data was investigated in [11,12], where the Multi-Level Context VoteNet was used to recognize 3D objects based on three context modules in the voting and classification stages of VoteNet. FCAF3D is a fully convolutional, anchor-free, indoor 3D object detection method based on a GSDN-like sparse convolutional network. It uses a voxel representation of a point cloud and processes voxels with sparse convolutions. The neural network accepts RGB-colored points and outputs a set of 3D object bounding boxes.

There are some other methods which, like ours, detect bounding boxes in 3D images. In [13], a method for cuboid detection and tracking using a multi RGB-D cameras is presented. This approach is designed for applications in the manufacturing industry, and detects boxes and other cuboid objects. Similarly to our method, this method also deals with rectangular geometric structures and returns 3D bounding boxes, but instead of detecting cuboid objects, our method detects cuboid empty space. The authors of [14] proposed an approach, based on 2D object detection, for 3D objects bounding box detection—without training. This method does not detect storage space.

### 2.1. Indoor Scenes and Furniture Synthesizing

In order to apply a deep learning approach, which requires large annotated datasets for training, we address in this work the generation of synthetic scenes and the automatic annotation of the bounding boxes of the storage space in these scenes. In [15], the authors created a large synthetic dataset of indoor 3D scenes with dense per-pixel annotations. In [16], a physics engine was used to randomly generate scenes from metrically scaled ShapeNet [1] objects. Finally, they obtained rendered RGB-D scenes and their ground truth annotation in a large video dataset. The authors of [17] manually created their own large dataset of synthetic 3D scenes with dense occupancy and semantic annotations. These 3D scenes contain 84 object categories, including furniture to which clutter was added to the shelves under supervision. In [18], ATISS, an autoregressive transformer architecture for creating indoor environments, based on room type and floor type, was presented. The input to the proposed model is a collection of 3D labeled bounding boxes of the scene objects with their corresponding room shapes. The output is the realistic furnished room with objects contextually belonging in a certain room type. However, these datasets were not suitable for network training in our research because they do not contain information about the storage space bounding boxes. We provide a dataset containing models and depth images of storage furniture with associated bounding boxes of the storage space, which was used for neural network training. This dataset was obtained by the proposed algorithm for automatic detection of empty spaces on voxelized 3D models, which does not require training. The whole process presented in our work is fully automatic and does not require any manual labeling.

The idea of creating assistive robots is presented in [19], where a simulation platform, Habitat 2.0 for training virtual robots in interactive 3D environment with rich physics, is presented. The authors presented the interactive dataset ReplicaCAD, consisting of apartment layouts with dynamical parameters of objects, their semantic classes, collision proxies, etc. However, there are no annotations of storage volumes, only surface annotations. In this simulator, robots can rearrange objects from a given list.

### 2.2. Finding Empty Space within Furniture

In [20], modular furniture is segmented in 2D image into doors, drawers, and shelves. The method returns 2D bounding boxes of the front faces of the closed furniture and does not provide information about volumes and 3D positions of storage space.

In our previous work [21], an algorithm is proposed for detecting empty spaces in storage furniture based on a voxelized 3D model. This algorithm takes voxelized 3D CAD models of furniture as input and computes 3D bounding boxes of the empty space within these models. We have improved this algorithm for better detection of storage volumes in more complex CAD models. While in [21], the proposed approach was tested only on few synthetic examples, in this study we used the improved method to generate a dataset for training of a neural network which detects storage space in depth images acquired by a real 3D camera.

There are not many other approaches that focus on empty space detection in storage furniture. In [17], related research is presented that provides information about empty space in furniture. The goal of the presented network, SSCNet, is to predict both occupancy and object category for voxels on the observed and occluded regions. This includes labeling of voxels belonging to different objects and voxels representing empty space, which is not necessarily storage space. Similarly, in [22], real-time semantic mapping for indoor environment was proposed. Labeled voxels represent parts of the room, including furniture and empty space. Storage volumes are, however, not labeled. This approach is evaluated on real-world 3D scenes. In contrast, the main idea of the approach presented in our work was to detect storage space bounding boxes in depth images, independently of the small objects within that space. The presented detection of storage space bounding boxes with supporting horizontal surface can be applied in automatic item placing and retrieval planning.

### 2.3. On-Shelf Availability and Object Placement

The problem of determining whether a product is available or out of stock on a retail shelf (OOS) is called on-shelf availability (OSA). In [23], the retail product detection problem is decomposed into a detection phase and a recognition phase. They segment the image into product and non-product segments and propose a method for determining shelf boundaries. Based on the peaks of a sample histogram, the segmentation of the shelves becomes clear to the human eye. They refer to the peaks as the positions of the products and their midpoints as the shelf boundaries. A 3D vision-based shelf monitoring (3D-VSM) system which automatically estimates OSA of products in a retail is proposed in [24,25]. In the proposed method, the reference model of the shelf is compared with a current depth image of the shelf, and alerts for OOS events are generated without a priori knowledge of the product type. This system was experimentally used in a retail store, where it accurately estimated OSA of the products. Another recent approach for determination of the OSA is given in [26]. The authors propose a machine learning pipeline for real-time empty shelf detection in 2D images for the purposes of retail performance improvement. Automatic detection of empty space between displayed products in retail on 2D images is proposed in [27]. The authors also provide an annotated dataset. In [28], a supervised learning approach is proposed for detecting out-of-stock items using texture, color, and geometry features in high-resolution panoramic images of supermarket shelves. Using images from a surveillance camera, a method for robust shelf monitoring through detection and classification is proposed in [29]. The method presented in [30] learns how to arrange objects to organize a shelf or sort objects in boxes according to users’ preferences. The main idea is that the robot has to assign the objects on the table to the shelves according to the predicted preferences of the user. However, this method does not calculate the available space on the shelves or in the boxes, but focuses more on assigning an object to the correct shelf. The above algorithms use 2D objects or images without depth to detect and identify shelves. In our work, we studied the detection of shelves, tables, or other furniture in depth images. Our method can also be used for a more general purpose, namely, to find storage space in previously unseen furniture, and it provides information about storage capacity whether it is currently occupied or not.

The result of our method could possibly be used with the planning algorithms proposed in [6,31]. A push planning algorithm proposed in [6] selects a sequence of actions to create space for the placement of objects. The objects have a shape, position, and orientation in 2D. The height of an object and the height of the storage volume are not considered. On the contrary, in the scene generation algorithm proposed in our paper, the height of the storage space bounding box and the heights of the objects are taken into account when placing objects on the shelves, that is, when creating clutter, and in the collision avoidance of objects in 3D. A data-driven method described in [31] was developed for arranging objects that automatically fill empty spaces on shelves with a single exampler to create arrangements. Their method creates a variety of style preserving arrangements that scale to different cabinet sizes and confirm style preservation by the user. Our algorithm could be used to detect storage volumes to which the referenced object placement method could be applied.

## 3. Methods

Several aspects must be considered when creating an appropriate training dataset for machine learning. First, it must be determined how large the training dataset must be for satisfactory neural network performance. There are some common estimates for the required number of scenes in the dataset, but it is impossible to know exactly how large the dataset needs to be. Second, the dataset should contain scenes which must reflect important characteristics of the robot’s work environment. Existing datasets of 3D indoor scenes either have a predefined clutter that cannot change, or they are clutter-free. Consequently, training a neural network on such a dataset would limit its application to that type of environment. Third, the characteristics of the robot visual sensor should be taken into account by simulating camera imperfections.

The solution to these problems is the development of a method which automatically generates realistic, scalable datasets that can be easily expanded and contain arbitrary type and amount of clutter. Datasets, which consist of scenes captured by real visual sensors, cannot be easily extended, as this would require the acquisition of additional scenes and their manual annotation, which is time-consuming and impractical. In contrast, new synthetic scenes can be generated relatively easily, making them more suitable for a scalable dataset. Due to the large number of publicly available 3D models of various classes, synthetic datasets also provide the ability to quickly change clutter in scenes, making it easy to change the context of the scene. However, synthetic images differ from images acquired by a real robot sensor.

The first part of the solution consists of a novel method that automatically detects empty storage volumes in 3D furniture models. For a given 3D model of a piece of furniture, this method provides one or multiple 3D bounding boxes, each representing a storage volume. The second part of our solution is a method which generates realistic scenes using 3D furniture models and adds clutter within the storage volumes under controlled conditions. Controlled conditions means that the user can select categories of objects to be used as clutter and specify their size and number. To create a realistic scene, 3D furniture models are placed on a floor, and some models are placed against a wall. The final step of this process is to render images from the created scenes. Simulated camera noise is superposed to the rendered depth images in order to obtain images similar to those captured by a real visual sensor. The described process is summarized in Algorithm 1. Inputs to the algorithm are a CAD model of empty furniture $M_{CAD}$ and a set of CAD models *clutter set* to be used to generate clutter, and ouputs are furniture model depth image $I_{3D}$ with set of annotated oriented bounding boxes of storage volumes OBB. The first part of the Algorithm 1 process consists of “voxelization” and “empty storage volume detection” procedures which are further explained in Section 3.1. The second part is the procedure “synthetic scene generator,” described in Section 3.2. The proposed training set generation method was tested by training a neural network and applying the trained network to detect storage volumes in depth images captured by a real 3D camera. An overview of the overall testing procedure is shown in Figure 1.
**Algorithm 1:** Synthetic dataset generator.**Input:** CAD models of empty furniture $M_{CAD}$ and clutterset**Output:** furniture model depth image $I_{3D}$ with set of annotated oriented bounding boxes of storage volumes OBB**procedure** Voxelization($M_{CAD}$)▹ Process I    Transform $M_{CAD}$ from mesh to voxel sapce    **return** $M_{voxel}$**end procedure**

**procedure** Empty storage volume detection($M_{voxel}$)▹ Process I    Step 1. Compute occupancy grid    Step 2. Compute occupancy distances    Step 3. Detect free space cores    Step 4. Detect empty storage volumes ESV    Step 5. Prune empty storage volumes    Step 6. Transform empty storage volumes from voxel to mesh space    Step 7. Adjust detected axis-aligned bounding boundig boxes AABB to enclosed storage volumes    **return** AABB**end procedure**

**procedure** Synthetic scene generator($M_{CAD},\;AABB,\;clutter\;set$)▹ Process II    Step 1. Scale $M_{CAD}$ using a random scale    Step 2. Add random items from clutterset to each AABB    Step 3. Place $M_{CAD}$ on a horizontal plane representing the floor    Step 4. With P(wall) generate vertical plane behind $M_{CAD}$ representing a wall    Step 5. Position virtual camera at $T_{PoV}$ facing $M_{CAD}$    Step 6. Render depth images $I_{3D}$    Step 7. Superpose simulated camera noise to $I_{3D}$    Step 8. Get oriented bounding boxes OBB from AABB and $C_{PoV}$    **return** $I_{3D},\;OBB$**end procedure**

### 3.1. Empty Storage Volume Detection Algorithm

The empty storage volume detection algorithm provides ground truth annotation of storage volumes in empty 3D furniture models. Furniture models are assumed to be in form of a triangular mesh, rectangular in shape with edges aligned with the axes of the mesh coordinate system. In general, empty storage volume can be considered as unoccupied space within or on top of furniture supported by a hard, flat, horizontal surface, referred to in this paper as a *supporting surface*. A supporting surface must be a part of the furniture, which is presumably rectangular in shape. Therefore, the ground truth annotations are cuboids enclosing empty volumes inside or on the top of furniture.

The 3D mesh models are rasterized into a n×n×n binary 3D voxel grid, which is used as input to the algorithm. The voxelized form allows straight-forward voxel classification into voxels occupied by furniture structure and unoccupied voxels. Voxels that are not occupied by the furniture structure indicate free space that could potentially be used as empty storage volume. In addition, it is possible to detect groups of unoccupied voxels above a layer of occupied voxels, which indicate storage volumes above a supporting surface. Based on this idea, we propose an empty storage volume detection algorithm consisting of four steps:Creating voxelized representation of the input meshDetection of free space;Detection of empty storage volume candidates in the free space;Empty storage volume candidate pruningBounding box fine-tunning.

Each of these steps is further explained below.

#### 3.1.1. Free Space Core Detection

The first step of the proposed algorithm is to create a voxelized reprezentation of the input 3D mesh. A voxel grid with dimensions n×n×n voxels is defined on the bounding cube of the input 3D mesh. Each voxel in this grid is assigned a binary value. The unoccupied voxels are assigned the value “1”, and are referred to as *free voxels*. Voxels occupied by the furniture structure are assigned the value “0”, and they are referred to as *occupied voxels*. This binary voxel grid is called the *occupancy grid*. In this section, *v* denotes the position of a voxel in the voxel grid, where $v=\left(x,y,z\right)\in R^{3}$ with the components *x*, *y*, and *z*, which represent the coordinates of the voxel in the voxel grid. Preforming the 3D chessboard distance transform (CDT) [32] of the voxel grid assigns to every free voxel *v* the chessboard distance to the closest occupied voxel *v’* [21]. The chessboard distance between two voxels *v* and *v’* is defined by
$$d\left(v,v^{\prime }\right)=\max(|x^{\prime }-x|,|y^{\prime }-y|,|z^{\prime }-z\left|\right)$$
value *D*(*v*) assigned to voxel *v* by the CDT is defined by
$$D\left(v\right)=\min_{v^{\prime }\in V_{O}}\left(d\left(v,v^{\prime }\right)\right)$$
where $V_{0}$ is the set of all occupied voxels. This value is referred to as *occupancy distance*.

Connected subsets of voxels corresponding to local maxima of the CDT with equal occupancy distances δ indicate the centers of empty volumes that can potentially be used as storage volumes. To isolate such groups of voxels, the algorithm determines local maxima of CDT in the 3×3×3 neighborhood of each voxel and creates a new binary grid. Voxels corresponding to the local maxima of the CDT are assigned the value “1”, while all other voxels are assigned the value “0”. The local maximum voxels are grouped using the connected component labeling. The groups of voxels obtained this way are referred to in this paper as *free space cores* (FSC). An example of a FSC is shown in Figure 2a,b.

Dilating a free space core by δ, i.e., using a cube of dimensions s×s×s, where s=2·δ+1, as the structuring element, yields an empty volume, which can be considered as an empty storage volume candidate. The proposed algorithm generates one storage volume candidate from one free space core. However, free space cores do not necessarily have to be cuboidal; rather, they resemble the shape of the environment they were detected in. Figure 2a shows an example of storage furniture with a non-cuboid storage space. This model will be used for further explanation.

The problem is that if the free space cores are not cuboidal, neither is the storage volume obtained by dilatation, which was the original assumption. Therefore, further processing is required. The proposed algorithm is based on the assumption that storage volumes are axis-aligned cuboids. Consequently, the free space cores must be adjusted to a rectangular shape so that the described dilation procedure can be used.

#### 3.1.2. Detection of Empty Storage Volume Candidates

Free space cores may be located inside a furniture model between its supporting surfaces or placed around the furniture. Cores in the furniture environment can be placed above, below, or next to the furniture model. The cores that are next to or below the furniture model have no required supporting surface, as defined in Section 3.1.1. Therefore, they are not considered for empty storage volume detection.

In an attempt to find the best axis-aligned cuboid that can be classified as an empty storage volume based on dilated free space core, the algorithm searches for the largest suitable volume per voxel grid axis. These volumes are referred to as storage volume candidates. Finally, the largest of the three proposed storage volume candidates is classified as a storage volume.

A connected set of voxels with the same *x*, *y*, or *z*-coordinate is referred to in this paper as a layer. A layer in a particular coordinate axis direction consists of voxels which have the same coordinate in this direction. An example is the layer in *y*-direction shown in Figure 2c. To find the largest storage volume per axis, the algorithm analyses the free space core voxel layers individually in the direction of each axis. Additionally, the algorithm adjusts the free space core to a rectangular shape by computing its bounding box. Figure 2a,b show free space core (orange voxels) consisting of 10 layers in the *x*-axis direction, 3 layers in the *y*-axis direction, and 7 layers in the *z*-axis direction. Figure 2c shows the extracted one free space core layer in *y*-axis direction. Note that the extracted free space core layers is not rectangular. Figure 2d shows free space core layer adjusted to a rectangular shape by computing its bounding box.

When a free space core layer is extracted, the algorithm checks if that layer has only one connected component. If multiple connected components are detected, the algorithm skips this layer and proceeds to the next layer because a decomposed free space core layer could possibly result in redundant and overlapping storage volumes.

The next step is to obtain a cuboidal volume based on the acquired free space core layer that can be proposed as a storage volume candidate. The free space core layer is dilated by the shortest distance to the nearest occupied voxels δ, as described in Section 3.1.1. An example is shown in Figure 3. Note that the obtained storage volume candidate can in some cases contain occupied voxels. This is corrected by the fine-tunning step explained in Section 3.1.4. The storage volume candidate can be classified as an empty storage volume if:(1)it has a sufficient size for storing the objects of interests, and(2)it has a hard flat horizontal surface underneath.

**Figure 3 sensors-22-06774-f003:**
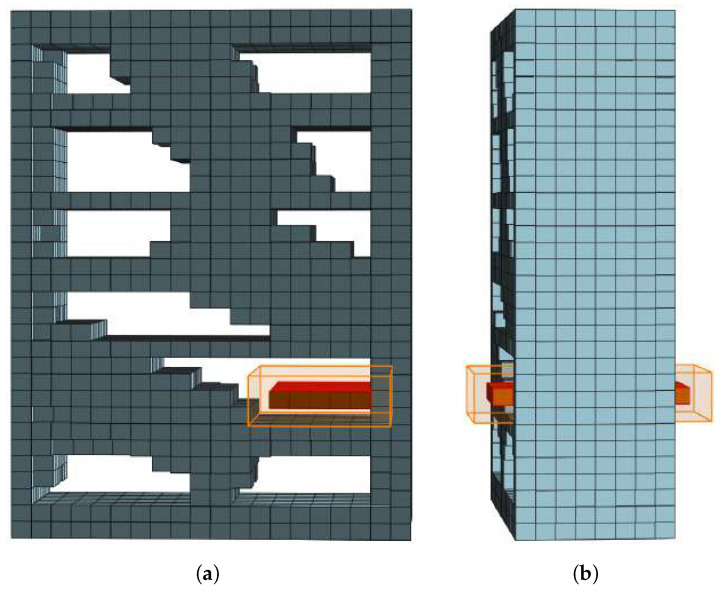
Rectangular free space core dilated by δ yields an empty storage volume candidate (orange bounding box): (**a**) Front view, (**b**) Side view.

The minimum size of the storage volume candidate $d_{sv}$ is defined in the mesh space, i.e., the space spanned by the mesh reference frame. Since storage volume candidates are created based on a voxelized model in the voxel space, their minimum size must be specified in voxels instead of meters. The minimal size of a storage volume candidate in voxels is determined as a cube with the sides of:$$d_{bb}=\lceil \frac{d_{sv}}{d_{v}}\rceil$$
where $d_{v}$ is the size of one voxel in the mesh space. Storage volume candidates of size greater or equal to $d_{bb}$ in all directions are further processed to determine their supporting surfaces, and those which do not satisfy this condition are rejected.

In general, the storage volume candidates generated by the described procedure are not necessarily completely supported by occupied voxels, as shown in Figure 3b. Thus, the algorithm must determine the supporting surface below each storage volume candidate and crop the storage volume candidate to fit its supporting surface.

To find the supporting surface of a storage volume candidate, the algorithm observes a layer of voxels in the occupancy grid directly below the storage volume candidate, referred to as the *supporting layer*. Only occupied voxels in the supporting layer can be used as supporting surfaces. Given the initial assumption that the storage volumes are cuboids, the supporting surfaces of these volumes are expected to be rectangular. Therefore, the problem of detecting supporting surface can be approached analogously to obtain free space cores.

The supporting layer can essentially be regarded as a 2D grid, where each cell of this grid represents one voxel. The cells are assigned binary values: “1” if the cell represents an occupied voxel below the storage volume candidate, or “0” otherwise. This representation allows detection of rectangular supporting planes.

A storage volume candidate can only be supported by one supporting surface. Therefore, it is necessary to check whether its supporting layer consists of a single connected component. If the supporting layer has multiple connected components, this storage volume candidate is eliminated from further process. The remaining storage volume candidates must be adjusted to the sizes and shapes of their supporting surfaces. To obtain a supporting surface that can be used as a storage surface, the algorithm performs a 2D CDT in the supporting layer. As the result, each cell of the supporting layer is assigned its occupancy distance. By identifying cells corresponding to occupancy distance local maxima and grouping them into connected subsets, *storage surface cores* are obtained.

Finally, the storage surface core is dilated by the occupancy distance δ′ of its cells, creating a bounding box of the usable *storage surface*. By adding the height of its storage volume candidate to the storage surface, the algorithm constructs a bounding box of an empty volume, as shown in Figure 4. If this bounding box is the largest empty storage volume candidate for a particular coordinate axis, it is classified as the storage volume candidate for that axis. The described process of storage volume candidate detection for each axis is summarized in Algorithm 2. After proposing three empty storage volume candidates for a given FSC (one for each axis), the algorithm classifies the largest of them as a storage volume.
**Algorithm 2:** An algorithm for detection of usable empty storage volumes in the occupancy grid.     **Input: free space core**     **Output: set of storage volume candidates** ***C***C←emptyset**for** each free space cores **do**    C′←emptyset    **for** each axis **do**        **for** each FSC layer **do**           **if** FSClayer has only one connected component **then**               **if** FSClayer is not rectangularly shaped **then**                   substitute FSC layer by its bounding box               **end if**               compute storage volume candidate by dilating FSC layer by δ               **if** minimum size of storage volume candidate ≥dbb **then**                   detect storage surface                   adjust storage volume candidate to storage surface                   **if** this is the largest storage volume candidate for the current axis **then**                       add storage volume candidate to C′                   **end if**               **end if**           **end if**        **end for**    **end for**    add the largest storage volume candidate contained in C′ to set *C***end for**

#### 3.1.3. Empty Storage Volume Candidate Pruning

Depending on the resolution of the rasterization of the mesh model to the voxels and the smoothness of the CAD models, some irregularities may appear in the voxelized model. Uneven furniture surfaces resulting from the voxelization process seemingly divide the free space into several segments. That leads to oversegmentation of the actual free space core into multiple connected components, referred to as *free space core segments*, ultimately resulting in multiple empty storage volumes instead of one.

To avoid this, the algorithm performs pruning of the empty storage volume candidates. Free space core segments representing the same empty space result in multiple storage volume candidates located above the same supporting surface. However, due to the rasterization problem described above, it is possible that these storage volumes have height differences of one voxel. Therefore, the algorithm analyzes sets of bounding boxes that have the supporting surface at the same height with a deviation of one voxel allowed, referred to as *neighboring height*. The storage volume candidates grouped by the neighboring height are then compared in pairs. If one of the storage volume candidates is completely contained within another, the algorithm prunes the inner candidate.

#### 3.1.4. Bounding Box Fine-Tuning

After detecting storage volumes in the voxel space, the algorithm transforms them to the mesh space. To transform the voxel space bounding box into the mesh space, it must be scaled by the voxelization factor $d_{v}$ and translated by the offset that occurs due to the voxelization process. Figure 5 shows the previously considered example of a voxel space bounding box in mesh space.

Since the purpose of the proposed automated storage volume detection is to create a large dataset for training neural networks in general, the metric bounding boxes should be adapted for this purpose. Most neural networks require ground truth labels that include some points of the object that should be recognized. On the other hand, we want to detect storage volumes which represent empty space. Nevertheless, storage volumes placed inside furniture are enclosed by the supporting surface, furniture walls, and upper shelves or furniture ceilings. The uppermost storage volumes are limited only by the supporting surface and the height of another object above it. Thus, to make an empty storage volume, an object that can be detected by neural network, the 3D bounding box must be adjusted to contain points on the surfaces enclosing this storage volume. This adjusted storage volume is referred to as *enclosed storage volume*, and the original bounding box from which it is created is referred to as the *detected bounding box*.

To obtain the enclosed storage volume, the algorithm must adjust the bounding box to the mesh vertices that belong to the nearest furniture wall and/or storage surface.

Due to the transformation from voxel to mesh space, the mesh vertices enclosing a storage volume are located in the neighboring region of $\pm d_{v}$ around the sides of the detected bounding box. That is, dilating and contracting the detected bounding box by $d_{v}$ creates a narrow region containing the mesh vertices of the furniture structure, as shown in Figure 6. The dilated mesh space bounding box is depicted in green, and the contracted mesh space bounding box is depicted in blue. The highlighted region represents the space in which the detected storage volume (orange) should be adjusted to the enclosed storage volume. In general, the extent of a storage volume is characterized by its axis-aligned bounding box defined by its minimum and maximum coordinates. Therefore, the minimum *x*-coordinate of the enclosed storage volume is $x_{min}^{{}^{\prime }}\in \left[x_{min}-d_{v},x_{min}+d_{v}\right]$ and the maximum point is $x_{max}^{{}^{\prime }}\in \left[x_{max}-d_{v},x_{max}+d_{v}\right]$. The minimum and the maximum bounding box coordinates in y and z directions can be determined analogously. The set of these six intervals containing the minimum and maximum bounding box coordinates for each axis is referred to as *storage volume range*.

A storage volume bounding box is adjusted by finding the peaks of the distribution of the mesh vertices in storage volume range. The histograms of the mesh model vertices were computed for each interval in the storage volume range. By extracting mesh vertices located in the storage volume range and creating a histogram of vertex coordinates for each axis, the algorithm retrieves six histograms. This is illustrated in Figure 7.

The storage volume range intervals define the lower and upper limits of the histogram bins. The number of histogram bins is determined as follows:$$n_{bins}=\lceil \left|\frac{r_{max}-r_{min}}{w_{bin}}\right|\rceil$$
where $w_{bin}$ is bin width determined empirically depending on the accuracy one wants to achieve; $r_{min}$ and $r_{max}$ are the minimal and maximal values defining the histogram range.

In the next step, the algorithm locates the peak value of each histogram. Using these peak values, the new minimum and maximum extents of the bounding box can be determined. To distinguish the flat histogram from histograms with apparent peaks, the algorithm defines the peak *H* as:$$H>\tau \;\cdot \;\bar{h}$$
where $\bar{h}$ is the mean histogram frequency and τ is an empirically determined threshold factor.

However, finding the peak of each histogram alone does not necessarily correspond to finding the furniture structure that encloses the observed empty storage volume. There are three main cases where this is not possible:the histogram is empty;there are no distinguishable peaks in the histogram;there are several peaks in the histogram.

The first case occurs when the considered empty storage volume is not completely enclosed by the furniture structure. In this case, the storage volume has no furniture structure above. That is, the storage volume is located on top of the furniture. An example is the table shown in Figure 8a. In this case, the algorithm uses the height of the detected bounding box as the height of the enclosed bounding box.

The histogram has no distinguishable peaks if the axis for which the histogram was calculated is aligned with a storage surface that is not bounded with furniture side-walls in the storage volume range; e.g., Figure 7a,c. The left-hand histogram in Figure 7a shows the histogram calculated in the direction aligned with a storage surface without furniture side-walls that is completely contained in the storage volume range. Figure 7c shows an example of a storage volume range that includes empty space and only part of the storage surface without furniture side-walls in the observed direction. It can be noted that the histograms have approximately uniform distribution of the mesh vertices along the supporting surface, and some histogram bins are empty. In both examples the algorithm will choose last non-empty histogram bin to adjust the size of the enclosed storage volume.

Multiple peaks in the histogram emerge when the storage volume is enclosed by furniture walls whose thickness is less than the histogram range. Two examples are the right histogram in Figure 7a and the bottom histogram in Figure 7b. In that case, the algorithm chooses the histogram peak that is closer to the center of the observed storage volume.

Finally, with the adjusted extents, the algorithm computes the axis-aligned bounding box as the last step in the process of empty storage volume ground truth annotation, which can be used for synthetic scene generation, explained in the next subsection. The considered example of bookshelf model with annotated enclosed storage volumes is shown in Figure 8c.

### 3.2. Synthetic Scenes Generation

In this section, we address the very specific task of training a neural network to predict bounding boxes of storage volumes. As far as we know, there is no publicly available dataset with storage volume annotations. Therefore, we designed a method which generates realistic synthetic datasets containing depth images of household furniture captured by a virtual camera. A depth image reproduces points from the point cloud that can be seen from a certain point of view, usually the point of view of the camera. A point cloud is a set of 3D points, where each point is assigned *x*, *y*, and *z* coordinates in Cartesian space. Each pixel of the depth image is assigned *x*, *y*, and *z* values, where *x* and *y* are image coordinates and *z* is the distance from the camera. In a real environment, 3D cameras provide depth images. In this work, during synthetic scene generation, depth images are rendered from the point clouds by positioning a virtual 3D camera on the 3D scene and capturing the depth image of the scene points from that viewpoint. The generated depth images are stored as point clouds with associated annotations of the bounding boxes representing storage space. These synthetic depth images resemble depth images captured by real 3D cameras. The detailed process of scene generation is explained below.

The input to this algorithm is a 3D mesh of a piece of furniture, commonly found in homes: a bookshelf, table, desk, or cabinet. Each mesh contains information about the bounding boxes of the storage units, provided by the empty storage volume detection algorithm explained in Section 3.1. Models in publicly available 3D model datasets are not necessarily provided in realistic sizes; e.g., models are often scaled to unit size. In that case, one way to scale models to realistic sizes is to utilize datasets of real objects with associated bounding boxes. An example of such a dataset is SUNRGB-D [33] containing real scenes with annotated bounding boxes of various pieces of furniture, where the height of the object matches the height of its associated bounding box. For the purposes of this research, we have created a database of realistic heights of the objects belonging to the considered household furniture classes taken from the SUNRGB-D dataset. Each 3D model of furniture was thus scaled by the ratio of randomly chosen realistic furniture instance height to the original height of the 3D model.

To create a realistic cluttered image, a predefined number of items were placed on the shelves of the furniture. Each item was placed in a random position on the shelf, in which it was not in the collision with other, already placed objects. These items are randomly selected 3D models of objects which are commonly placed on furniture, such as lamps, bottles, mugs, jars, and bowls or cuboids. Cuboid templates have a unit base and height of 0.6 m. Using the information about the bounding boxes of the storage volumes, the objects were placed at the bottoms of the bounding boxes which were aligned with the shelves, and their sizes were scaled by a random scaling factor drawn from a user-defined uniform distribution to fit within the storage volume. Since many objects found on the shelves in real scenarios are cuboids (books, boxes), the probability of cuboid objects being present on the shelves was set to 50%. Some examples of generated models of furniture with clutter on the shelves are shown in Figure 9a.

Some scenes include other furniture models—flower pots, lamps, sofas, and chairs—that are also commonly found in households but do not provide storage volumes. These negative samples in the training dataset reduce the amount of false positives during evaluation, thereby increasing the precision.

After scaling the storage furniture models and creating clutter on the models’ shelves, the depth images were rendered. A furniture model was placed on a horizontal plane representing the floor. Some storage furniture in real scenarios is often placed against the wall (e.g., bookshelves), whereas other furniture is not (e.g., dining tables). The wall was modeled as a vertical plane behind the furniture model. Depending on the user-defined probability P(wall) that the wall is behind the model, the wall was also rendered on the image.

A virtual camera was randomly positioned at $T_{PoV}$ within the boundaries, where it could be in real scenarios. Assuming that the robot moved around a room and places objects on the shelves, the camera was positioned between 1 and 2 m away from the furniture, between 1.2 and 1.5 m above the floor surface, and within 3 m to the left or right of the object center. The camera was aimed at the point within range of 20% of the object’s height above and below of the object’s center. In the end, simulated camera noise was applied to obtain images similar to those captured by the depth camera. Examples of depth images generated by the proposed method are shown in Figure 9b.

Finally, the rendered images were saved as point clouds, along with the associated information about the oriented bounding boxes of the storage volumes and votes generated by the procedure given in [2]. Note that the votes were required for the training of the VoteNet [2] neural network, used in this research. Examples of stored point clouds can be found in Figure 9c. On these images, the bounding boxes of the storage furniture are also visualized, and their edges are colored pink. In some cases, storage furniture is not completely visible from a randomly selected viewpoint. Since it is not expected that the detector recognizes a storage volume of which only a small part is visible in the image, the bounding boxes whose centers are not contained within the camera’s field of view were not assigned to the image. An example of such an image is shown in the first row of Figure 9. Note that the top shelf of the bookshelf model is not visible in the depth image Figure 9b, and therefore, its bounding box was not associated with this image. This can be seen in Figure 9c, where only shelves visible on the depth image have associated bounding boxes (colored pink).

## 4. Evaluation

The proposed algorithm for detection of empty storage volumes in 3D furniture models, described in Section 3.1, requires voxelized representations of these models. Therefore, a voxelization of 3D triangular mesh models was performed as a preprocessing step. This process required watertight meshes in order to be able to distinguish between the free space and space occupied by furniture. To generate watertight furniture meshes, we used the code described in [34], and voxelization was performed as in [35] using [36].

The proposed approach can be used to generate training datasets for neural networks which operate on 3D point clouds. We used the VoteNet neural network [2] and FCAF3D [3] to test our approach. The trained network was tested with both the syntetic evaluation dataset and depth images acquired with a real RGB-D camera.

Further explanation of the selected neural networks and detailed overview of the evaluation dataset are explained below.

### 4.1. Adaptation and Training of Neural Networks

VoteNet takes as input a 3D point cloud and returns as output a set of oriented 3D bounding boxes of the detected objects, where each bounding box is assigned probabilities of belonging to each of the considered object classes. This neural network consists of a backbone, a voting module, and a proposal module.

The VoteNet’s backbone feature learning network is based on PointNet++ [37], which has four set abstraction (SA) layers and two feature propagation (FP) layers. Each SA layer assigns a feature vector to a subset of points from the previous SA layer contained in a spherical receptive field with a given receptive radius. The SA layer also subsamples the input point cloud with the farthest point sampling. The SA layers have increasing receptive radii of 0.2, 0.4, 0.8, and 1.2 in meters, and they sub-sample the input into 2048, 1024, 512, and 256 points, respectively. Each feature propagation layer performs upsampling of the point features, where the output point features are computed as weighted averages of the three nearest input points’ features. In addition, it also combines the skip-linked features through a multi-layer perceptron (MLP), where the interpolated features and the skip-linked features are concatenated before being fed into the MLP. The two FP layers up-sample the 4th SA layer’s output back into 1024 points with 256-dim features and 3D coordinates. The output of the VoteNet’s backbone is a set of seed points with assigned feature vectors. Given a set of seed points, a shared voting module generates votes from each seed point independently.

The voting module is realized with an MLP with fully connected (FC) layers of sizes 256, 256, and 259; ReLU; and batch normalization, which computes the Euclidean space offset and a feature offset for each seed point. A vote is generated for each seed point by adding the computed offsets to the seed point’s coordinate vector and its feature vector. Then, vote clustering is performed by selecting a subset of K votes using the farthest point sampling and forming clusters of votes in a spherical neighborhood of each of these selected votes. For each cluster, an object proposal is generated by MLPs that compute a proposal vector consisting of an objectness score, bounding box parameters and classification scores for each target object class. The proposals are post-processed by 3D non-maximum suppression (NMS), resulting in the final object predictions.

The VoteNet was initially created for detection of ten classes in SUN RGB-D dataset: bed, table, sofa, chair, toilet, desk, dresser, night stand, bookshelf, and bathtub in depth images. Adapting VoteNet to detect only a single class—a storage volume—was straightforward: we simply used our training dataset, in which all storage volume bounding boxes had the same label. Since only this one class label occurs in the dataset, we effectively trained VoteNet to recognize only a single class. The network was trained end-to-end and from scratch with the Adam optimizer, batch size 8, and an initial learning rate of 0.001. Training the network on the proposed dataset of 7194 scenes for 180 epochs on NVIDIA GeForce GTX 1060 took around 32 h.

The FCAF3D neural network accepts a colored 3D point cloud as input and outputs a set of 3D object bounding boxes. Its architecture consists of a backbone, a neck, and a head. Similarly to GSDN [38], the sparse 3D modification of ResNet34 [39] named HDResNet34 [40] is used as the backbone. The backbone generates four levels of hierarchical sparse tensor feature maps $B_{l}$, l = 1, 2, 3, 4 from the feature maps of the previous level. The neck processes feature maps $B_{l}$ and generates tensors $N_{l}$ at each feature level *l*. The tensors $N_{l}$ are obtained by skip connections between tensor feature maps $B_{l}$ and upsampled tensors $N_{l+1}$. By upsampling the tensors $N_{l+1}$ with a sparse transposed 3D convolutional layer, the tensors $T_{l}$ used for skip connections are obtained. These tensors are added to the tensor feature maps $B_{l}$, and the obtained tensor is processed by a sparse 3D convolutional layer, resulting in the tensors $N_{l}$. The tensors $N_{l}$ are used as inputs to the FCAF3D head, which consists of three parallel sparse convolutional layers whose weights are shared across all feature levels. For each location, these layers output classification probabilities, bounding box regression parameters, and centerness [40]. The size of the output classification layer of the original FCAF is 10, since it is designed to detect 10 object classes. In our training we used only one object class—storage volume, the only class that occurs in our training dataset. During training, FCAF3D outputs locations for different feature levels that should be assigned to ground truth bounding boxes. For each bounding box, the last feature level at which this bounding box covers at least $N_{loc}$ locations is selected, where $N_{loc}$ is a hyperparameter. If there is no such feature level, the first level is selected. Locations are also filtered by center sampling [40], where only the points near the center of the bounding box are considered as positive matches. By assignment, some locations are matched with ground truth bounding boxes. These locations are associated with ground truth labels and centerness values. During inference, the predicted classification probabilities are multiplied by the centerness values, and the value obtained is used for non-maximum suppression.

In initial point cloud voxelization, the voxel size is set to 0.01 m and the number of points to 100,000. We changed the network configuration settings to take 50,000 points as input, and set the number of classes to one. The other settings were set to the default values. Training of FCAF3D on NVIDIA GeForce RTX 3080 takes approximately 7.5 h on the proposed dataset.

### 4.2. Evaluation on the Synthetic Dataset

A total of 9998 synthetic scenes with 34,727 annotated storage volumes were created. The dataset was split approximately 90:10 for training and testing; 20% of the training dataset was used for validation during training. The distribution of the dataset per furniture class with respect to each split of the dataset is shown in Table 1.

Table 2 shows the minimal, maximal, and median bounding box (according to the volume) for each furniture class in the synthetic dataset together with the corresponding size (length (*x*), height (*y*), and width (*z*)) of a particular bounding box. This table also shows the median length (*x*), height (*y*), and width (*z*) of each bounding box. Distribution of bounding box volumes size per furniture class is shown in Figure 10. The smallest dimension is 7 cm, and the largest is 4.86 m. The average size per axis of the storage volumes in the dataset is 0.31×0.43×0.31 m.

### 4.3. RGB-D Images of Real Scenes

To test the proposed storage volume detector in a real life scenario, scenes from two homes and two offices were captured with the Asus Xtion Pro Live RGB-D camera. The camera was placed at a height between 1.2 and 1.5 m and at a distance of 1 to 3 m from the storage furniture. In total, 151 RGB-D images were captured. The dataset was captured in rooms and offices containing furniture of the same type as the models used for storage volume detector training. This included a living room, family room, bedroom, pantry, kitchen, dining room, office, and library. In post-processing, the captured scenes were aligned with a dominant horizontal plane in the image.

A detailed overview of the dataset acquired with the RGB-D camera by number of captured scenes and number of annotated storage volumes is shown in Table 3. The scenes in this dataset are grouped by room type in which they were taken. Table 4 shows the minimal, maximal, and median bounding boxes (according to the volume) for each furniture class in the synthetic dataset together with the corresponding size (length (*x*), height (*y*), and width (*z*)) of a particular bounding box. This table also shows the median length (*x*), height (*y*), and width (*z*) of all bounding boxes.

### 4.4. Ground Truth Annotation

Annotation of storage volumes on the depth images of real scenes is one of the contributions made in this paper. The ground truth annotation algorithm provides the center, size, and heading angle of each bounding box.

The proposed annotation method requires a 3D triangular mesh of the scene created from a depth image acquired by an RGB-D camera. At least six points in a scene must be selected by the user to define the bounding box of a storage volume. The selected points are defined as follows: the first two points, $T_{1}$ and $T_{2}$, must lie on a horizontal edge of the storage volume. This edge is shared by two of six planes defining the storage volume. Each of the remaining points $T_{3},T_{4},\ldots ,T_{6}$ must lie in one of the remaining four planes of the storage space.

The first two points, $T_{1}$ and $T_{2}$, defined as $T_{1}=\left(x_{1},y_{1},z_{1}\right)$ and $T_{2}=\left(x_{2},y_{2},z_{2}\right)$, are used to define the *x*-axis of the coordinate system of the bounding box, defined by $x=\frac{\vec{T_{1}T_{2}}}{|\vec{T_{1}T_{2}}|}$. We assume that the *z*-axis is the vertical axis of the coordinate system of the scene. The orthogonality of the *x* and *z*-axes is assured by aligning the scene mesh with the vertical axis. Given *x* and *z*-axes, the *y*-axis is defined as y=z×x. The obtained *x*, *y*, and *z*-axes define the orientation of the bounding box coordinate system. The sides and edges of the bounding box are aligned with these axes. Rotation matrix ${}^{BB}R_{S}={\left[x\;y\;z\right]}^{T}$, where *x*, *y*, and *z* are unit vectors defined above, is a matrix for transforming coordinates of the selected points originally represented in the coordinate system of the scene (S) to the coordinate system of bounding box.

Let *X*, *Y*, and *Z* be the sets of the coordinates of the selected points with respect to the bounding box coordinate system defined as follows: $X=\{x_{1},x_{2},\ldots ,x_{n}\}$, $Y=\{y_{1},y_{2},\ldots ,y_{n}\}$ and $Z=\{z_{1},z_{2},\ldots ,z_{n}\}$, where *n* is number of the selected points. Let $x_{min}$, $x_{max}$, $y_{min}$, $y_{max}$, $z_{min}$, and $z_{max}$ be the minimum and maximum values of the sets *X*, *Y*, and *Z*, respectively.

The center of the bounding box is defined as
(1)$${}^{BB}C=\left(\frac{x_{min}+x_{max}}{2},\frac{y_{min}+y_{max}}{2},\frac{z_{min}+z_{max}}{2}\right).$$

The center of the bounding box represented with respect to the scene coordinate system is obtained by transforming the bounding box center computed by (Equation 1) using the rotation matrix ${}^{S}R_{BB}$.

The size of the bounding box, i.e., its length, width, and height, are defined by vector
$$s=(|x_{max}-x_{min}|,|y_{max}-y_{min}|,|z_{max}-z_{min}|).$$

The heading angle defining the orientation of the bounding box with respect to the scene coordinate system is defined as the angle α between the *x*-axis of the coordinate system of the scene and the *x*-axis of the coordinate system of the bounding box. The final result of the proposed algorithm is a set of bounding boxes, where each bounding box is defined by ${}^{BB}C,\;s$ and α.

Some storage volumes are defined by design of the furniture, such as storage volume inside of the shelf, as shown in Figure 8b. However, the top of some furniture, such as a cabinet or table, might be used for the disposal of items as well. Hence, the volume on the top of such furniture whose base is defined by its top surface can be considered as a storage volume. An example is given in Figure 8a. The height of such storage volumes is not defined by the design of the furniture but depends on how much space is between the top of the furniture and the ceiling. Furthermore, in some scenes the top of the furniture is outside of the camera’s field of view, so it is not visible in the image. In such cases, we define the heights of such storage volumes as the maximum heights of the items that are to be placed in storage volumes. In our evaluation experiments, this height was constant and amounted to 0.5 m. The option to choose whether the height of a storage volume is defined by the furniture design or the predefined height of 0.5 m should be used is left to the user. If the user selects the mode with the predefined fixed height, the sixth point is not selected by the user, but the algorithm generates it by adding the value 0.5 to the *z*-coordinate of the point $T_{1}$. The rest of the algorithm remains unchanged.

An example of depth image of a furniture and ground truth annotation of its storage volumes is shown in Figure 11.

## 5. Results

The storage space detectors, trained on VoteNet and FCAF3D neural networks, were evaluated with two test datasets: synthetically generated and real depth images. The evaluation was performed by computing the 3D volume intersection over the union (IoU) values of ground truth and prediction boxes, where thresholds 0.25 and 0.5 were used to calculate the average recall for generation and average precision for detection, as described in [2].

The proposals were post-processed using a 3D non-maximum suppression (NMS) IoU threshold of 0.1 and a confidence threshold of 0.75. The storage volume detector was tested on the synthetic dataset in two ways: on the entire evaluation dataset and on a sub-dataset consisting of a bookshelf and negative scenes. The training of the storage volume detector was performed on a synthetic dataset, half of which consisted of scenes with bookshelves, and the remaining scenes of the dataset were evenly distributed among the other classes. Therefore, the trained network was tested separately in scenes with bookshelves and in scenes with cabinets, desks, and tables. The results of this test, together with the testing results on the real dataset, can be found in Table 5.

Table 6 shows evaluation results of storage space detection on the real dataset per room where scenes were captured. Examples of qualitative results of storage volume detection in real images are shown in Figure 12, Figure 13 and Figure 14.

### 5.1. Synthetic Dataset

Since this is the first method proposed for generating training data for storage volume detection, it is not possible to compare its success to that of other methods which solve the same problem. Nevertheless, as some form of comparison, we can use the results obtained by the VoteNet trained for houshold object detection on a similar data type (indoor depth images) using the SUN RGB-D dataset [33], which achieved average precision (mAP) of 57.7% and 78.7% recall, with the 3D IoU threshold of 0.25.

However, to test the performance of our dataset, we trained two adapted neural networks, VoteNet and FCAF3D. Overall, the trained VoteNet achieved storage volume detection precision of 43.66% and recall 62.29%; and FCAF3D achieved precision of 47.72% and recall 71.41% on synthetic dataset with the IoU threshold of 0.25.

### 5.2. Real Dataset

The evaluation of the proposed storage space detection method on the real dataset was conducted in two setups. In the first testing setup, the evaluation was conducted for each environment in the dataset individually. The second setup was composed of all dataset scenes. An overview of results is given in Table 6. The highest precision and recall were achieved in scenes from Home 2—Kitchen, Office 1, and Office 2, which contain close-up scenes of a dining table, a bookshelf, and a cabinet. The lowest precision was obtained in scenes containing heterogeneous furniture captured from a greater distance, e.g., Family room in Home 1.

## 6. Discussion

The generated synthetic dataset was successfully used to train two neural networks adapted from indoor object detection to the storage space detection. This indicates that the combination of the proposed methods for automatically labeling empty storage volumes and generating synthetic scenes achieved its goal.

Comparing the precision results of the storage volume detectors in the synthetic dataset (Table 5), we found that the results for both detectors are similar at the required IoU of 0.25. The difference is notable at the required IoU of 0.5 in favor of the FCAF3D storage volume detector. For both IoU settings, the FCAF3D detector has significantly higher recall than VoteNet. Therefore, the detector trained on FCAF3D achieved better overall precision and recall results on the synthetic dataset. However, the detector trained on VoteNet achieved 18% higher precision on the real dataset, and FCAF3D has 8% higher recall. Since the purpose of the proposed method is to detect household furniture storage in real depth images, the results of VoteNet detector are further discussed.

Table 5 shows that the detection of storage volumes performs slightly better on the bookshelf class compared to other furniture classes. The visual results of storage volume detection in the synthetic dataset show similarity in the number, size, and position of predictions relative to the ground truth labels.

Furthermore, by observing the results obtained on the real dataset, and in particular comparing the scenes of the same bookshelf in two settings in room Office 1—an empty bookshelf and a bookshelf with a specific clutter—we can see that better results were achieved for a bookshelf with moderate clutter. Qualitative results obtained on the real dataset indicate that the storage volume detector achieves better results for images taken closer to the furniture. For example, the bookshelf in Figure 13a has three labeled ground truth storage volumes, and three similar volumes were detected by the storage volume detector (Figure 13b). When observing that bookshelf from a greater distance (Figure 13c,d), the storage volume detector found some false positive volumes and did not precisely match the detected true positives to the labeled ground truth. The same can be concluded for the bookshelf in Figure 14a–d. However, the false positive storage volumes in Figure 13f are mostly in the expected locations in the cabinet, but were not hand-labeled as ground truth because they are not visible enough to the human eye. This indicates that the storage volume detector was trained to generally detect storage volumes in any geometric structure that resembles the storage volumes described in Section 3.1. On the other hand, it is evident that the ground truth labeling largely depends on the human’s subjective interpretation of the scene. In the process of manual ground truth labeling of the real dataset, we had to decide whether partially visible storage volumes should be labeled. Labeling all potential storage volumes, including those that are barely visible, would result in a seemingly small recall. This is because it is unrealistic to expect a neural network to successfully recognize the storage volume in situations that are difficult to assess even by human. Still, as seen in Figure 13f, the neural network was able to detect a partially visible storage volumes which were not labeled, causing false positive predictions that reduced precision. For these reasons, we decided to label only those storage volumes that were at least 50% visible according to human judgment.

High recall was obtained for the scenes Kitchen from Home 2 and Cabinet from Office 2. This is because these scenes contain relatively small numbers of annotated storage volumes. All ground truth bounding boxes in these scenes were found, resulting in high recall, but some false positives also occurred (Figure 14f), resulting in somewhat low precision.

## 7. Conclusions

The goal of the presented research was to develop an algorithm that enables a service robot to detect storage space in a piece of furniture being viewed for the first time, even if that storage space is cluttered with various items. To the best of our knowledge, this is the first study to address this particular problem, which is of great importance for future-related robotic applications. A robot that is able to determine the an available storage space, and its location and size, could use this information to decide whether a particular item will fit in and plan actions to place that item in that storage space. Knowledge of the available storage capacity in the robot’s work environment would allow the robot to allocate a given set of items to the designated furniture.

Furthermore, we investigated the possibility of solving the considered problem using machine learning with minimal human effort. As a result of this research, we developed a method for automatically generating neural network training data that allow a neural network designed for detecting objects in depth images to detect storage volumes. The required human effort is reduced to the selection of 3D furniture models and the parameterization of the scene generation algorithm; manual annotation is not required. This is achieved by two methods, one for automatic detection of storage volumes in 3D furniture models and the other for automatic generation of realistic scenes containing these models with clutter on their shelves. The generated scenes with the associated bounding boxes of the storage volumes were used as the input dataset for training the VoteNet neural network.

The trained network was used to predict the storage space bounding boxes in a dataset of images captured by a real 3D vision sensor and achieved an overall precision of 42.22%, and a recall of 66.67% for an IoU of 0.25. Moreover, it can be concluded from the performed experiments that the neural network accurately estimates the orientation of the bounding boxes, i.e., aligns them well with the edges and sides of the real storage volumes. However, the large differences in precision and recall obtained for different IoU thresholds of 0.25 and 0.5 suggest that the detector finds storage volumes approximately where they actually are, but determines their boundaries rather inaccurately.

Since this was the first study on this specific problem, we consider the results obtained interesting. However, there is still much room for improvement. It would be worthwhile to investigate what performance improvement could be achieved with a larger training dataset containing more furniture models and scenes with multiple storage furniture models. In addition, the detection performance could likely be improved by creating scenes with a greater variety of background objects and objects positioned in the storage space.

As storage furniture is normally relatively large and is usually perceived from a short distance away, it is expected that such objects will often be only partially visible in the captured images during the regular operation of a service robot. Detection of partially visible storage volumes and appropriate evaluation criteria for the proposed solutions are interesting problems worth investigating.

Finally, in order to determine what accuracy of storage space parameters is required for practical use, some experiments with real robots picking up and placing items from and into the shelves should be conducted in the continuation of this research.

## Figures and Tables

**Figure 1 sensors-22-06774-f001:**
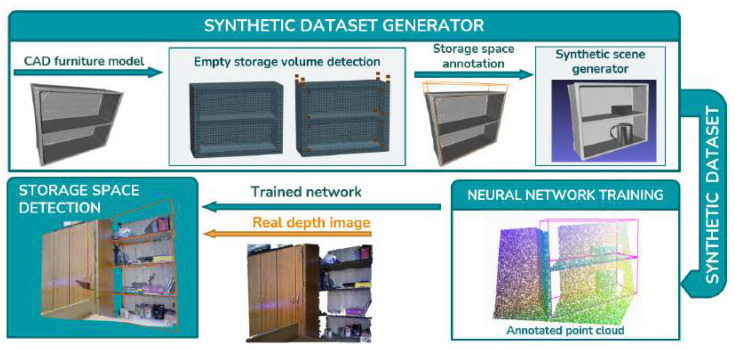
The method overview.

**Figure 2 sensors-22-06774-f002:**
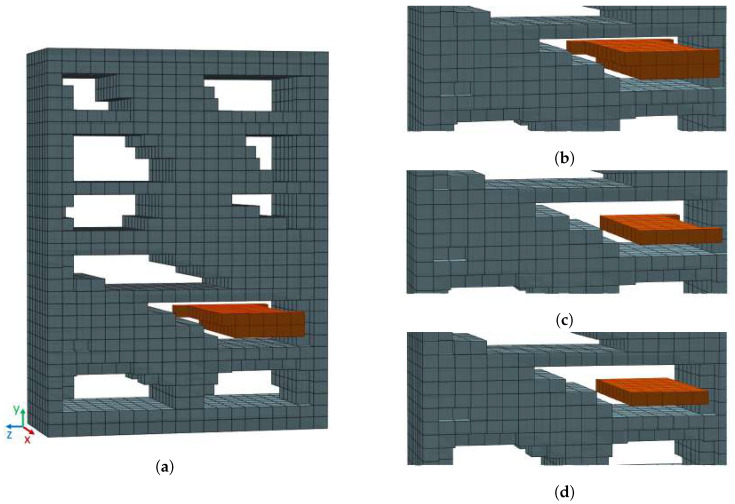
Example of voxelized storage furniture model (bookshelf) with detected free space core (FSC—orange voxels). (**a**) Front view of furniture model with selected FSC (**b**) FSC zoomed, (**c**) One FSC layer in *y*-axis direction, (**d**) FSC layer adjusted.

**Figure 4 sensors-22-06774-f004:**
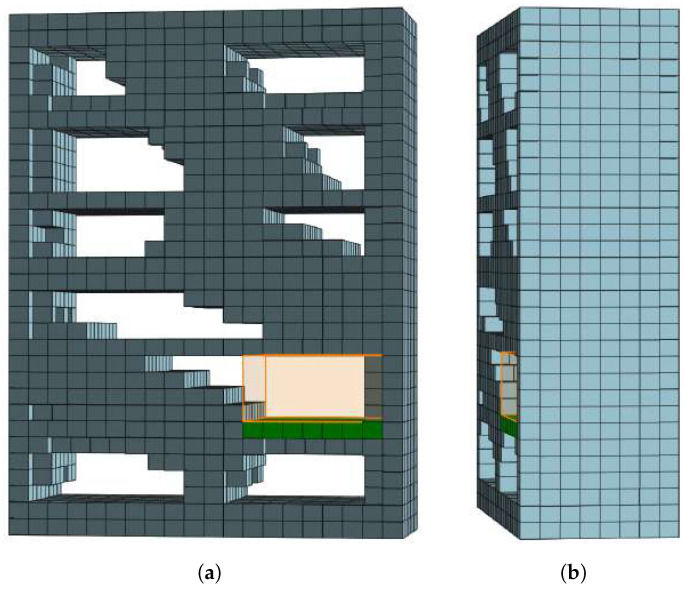
Detected storage surface (green voxels) of a FSC from Figure 3 and its corresponding storage volume (orange bounding box): (**a**) Front view, (**b**) Side view.

**Figure 5 sensors-22-06774-f005:**
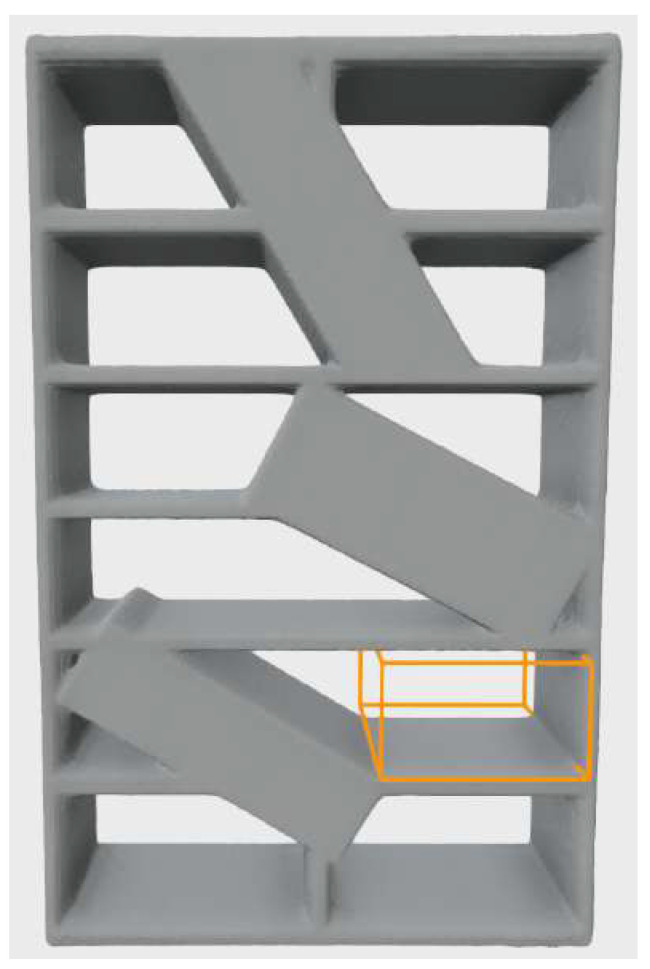
Selected mesh furniture model with detected storage volume in mesh space. The storage volume is depicted with an orange bounding box.

**Figure 6 sensors-22-06774-f006:**
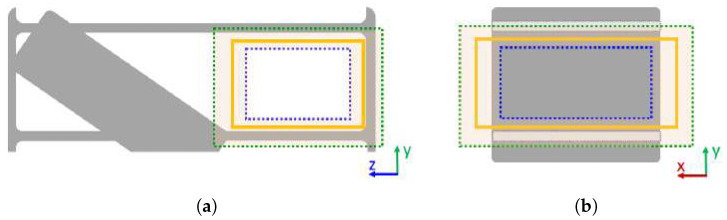
The detected bounding box is depicted with an orange rectangle. Dilating or contracting that bounding box by $d_{v}$ would create green or blue bounding boxes, respectively. The highlighted volume between the blue and green bounding boxes contains mesh vertices that enclose the observed storage volume: (**a**) Front view, (**b**) Side view.

**Figure 7 sensors-22-06774-f007:**
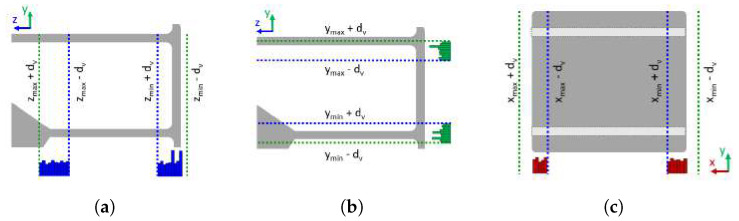
Mesh vertices distribution in storage volume range along: (**a**) *z*-axis (front view), (**b**) *y*-axis (front view), (**c**) *x*-axis (side view).

**Figure 8 sensors-22-06774-f008:**
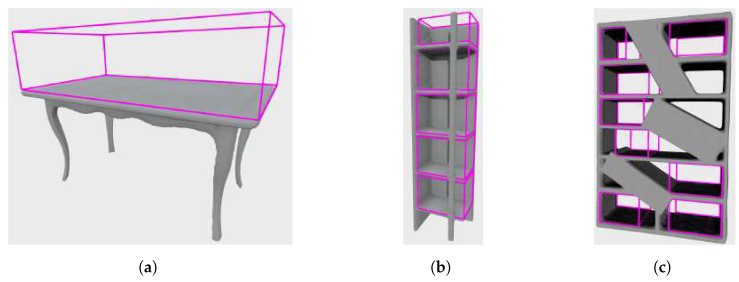
Examples of labeled ground truth annotations (purple bounding boxes) of storage volumes in selected furniture models: (**a**) Table, (**b**) Bookshelf A, (**c**) Bookshelf B.

**Figure 9 sensors-22-06774-f009:**
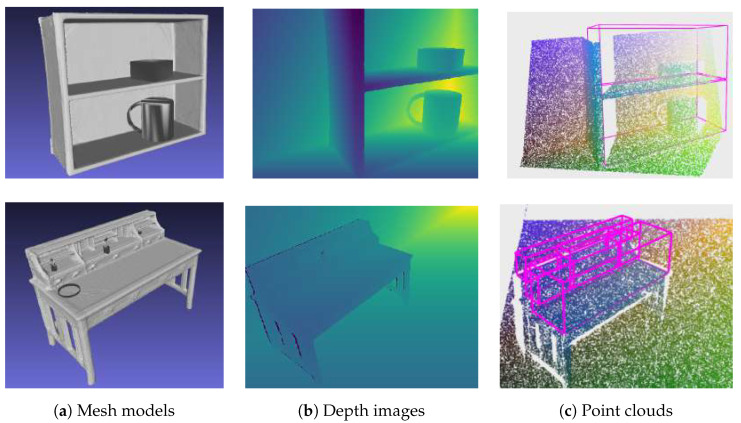
Scenes with furniture and items placed in the storage space: (**a**) Examples of generated cluttered scenes of a bookshelf and a table, (**b**) Generated depth images, (**c**) Generated point clouds with associated bounding boxes (purple colored edges).

**Figure 10 sensors-22-06774-f010:**
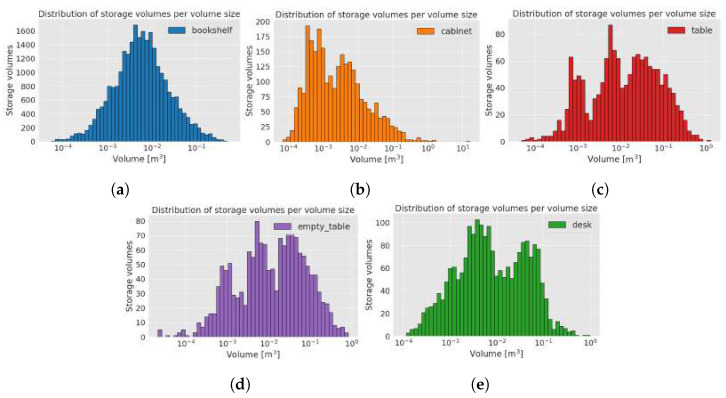
Storage volume per volume size for each furniture class: (**a**) bookshelf, (**b**) cabinet, (**c**) table, (**d**) empty table, and (**e**) desk.

**Figure 11 sensors-22-06774-f011:**
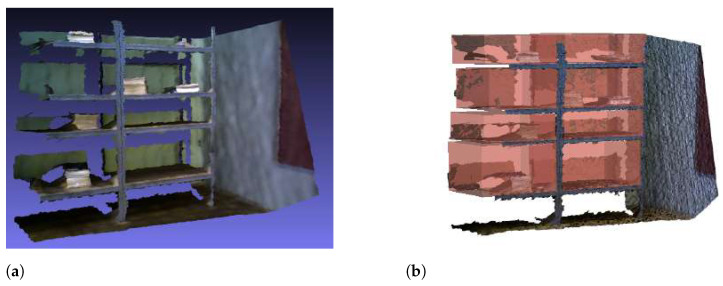
Ground truth annotation of the household furniture. (**a**) A bookshelf captured by an RGB-D camera. (**b**) A bookshelf with ground truth annotation.

**Figure 12 sensors-22-06774-f012:**
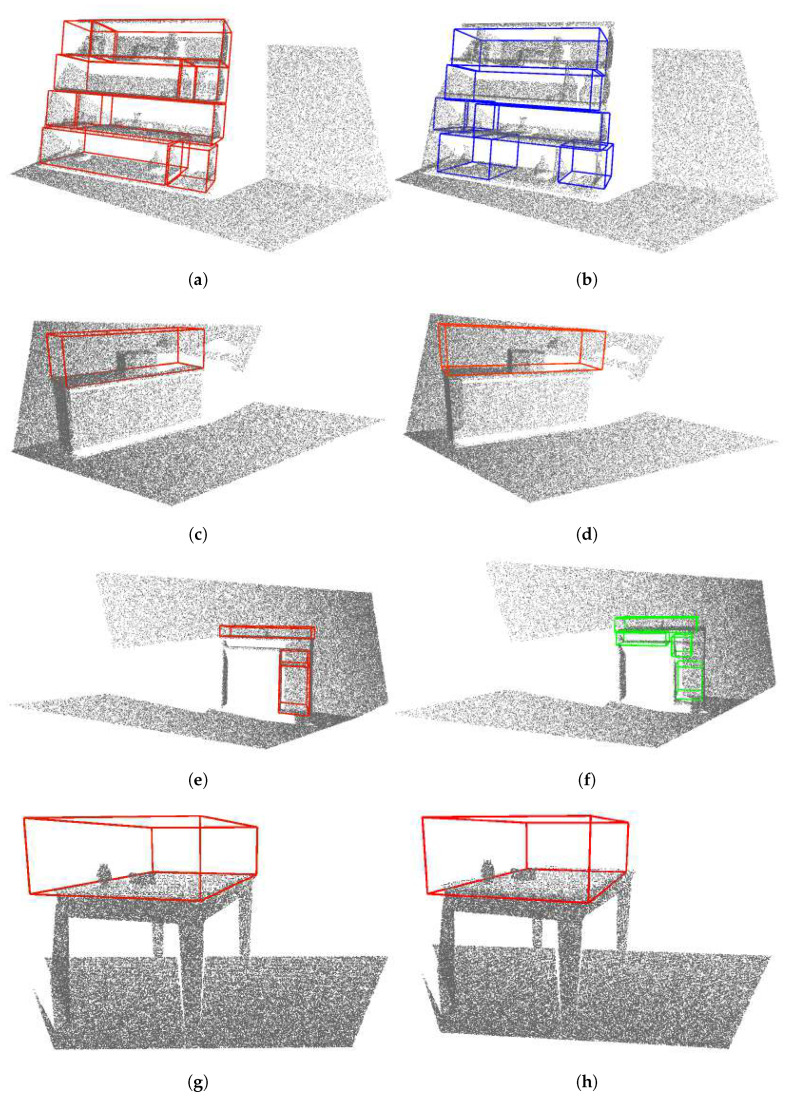
Examples of storage volumes in depth images for different furniture classes: (**a**,**b**) bookshelf, (**c**,**d**) cabinet, (**e**,**f**) desk, and (**g**,**h**) table. The left column shows ground truth annotation of storage volumes in the selected furniture classes. The right column shows the storage volumes predicted in these furniture models by the VoteNet detector.

**Figure 13 sensors-22-06774-f013:**
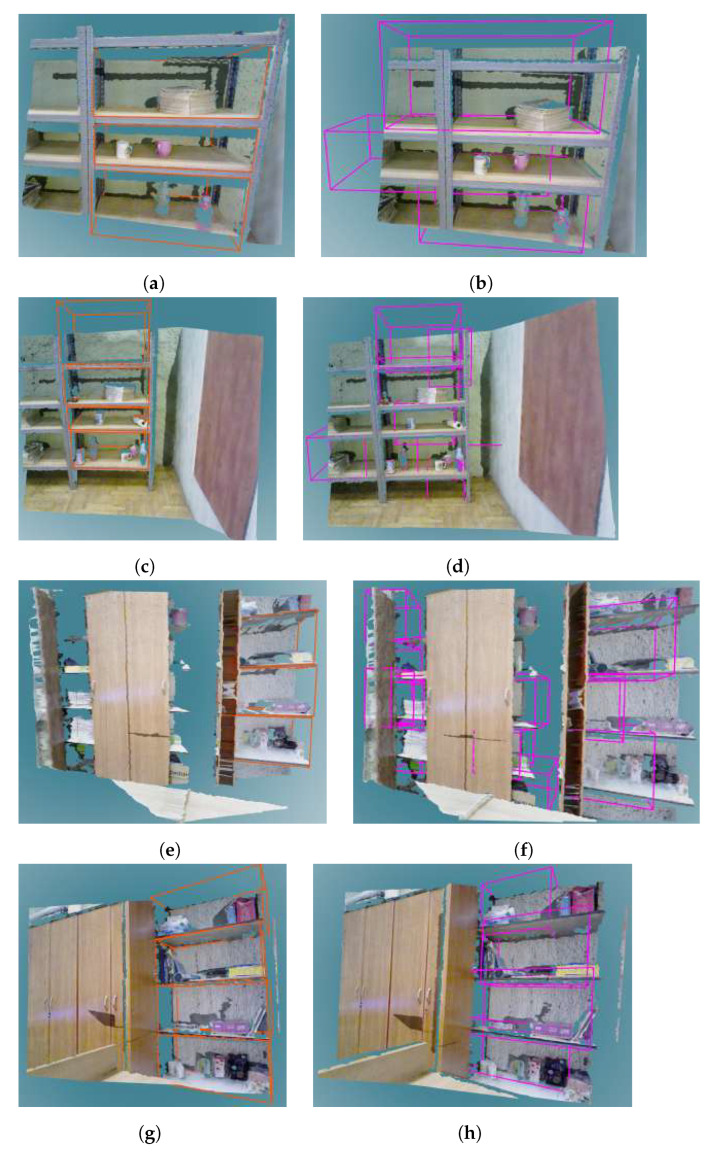
Examples of storage volumes in depth images contained in a real dataset. The left column shows ground truth annotation of storage volumes in the selected furniture classes. The right column shows the storage volumes predicted in these furniture models by the VoteNet detector. (**a**,**b**) Show a close-up view of a bookshelf, (**c**,**d**) Show a bookshelf seen from a distance, (**e**,**f**) Show a close-up view of a cabinet, (**g**,**h**) Show a cabinet seen from a distance.

**Figure 14 sensors-22-06774-f014:**
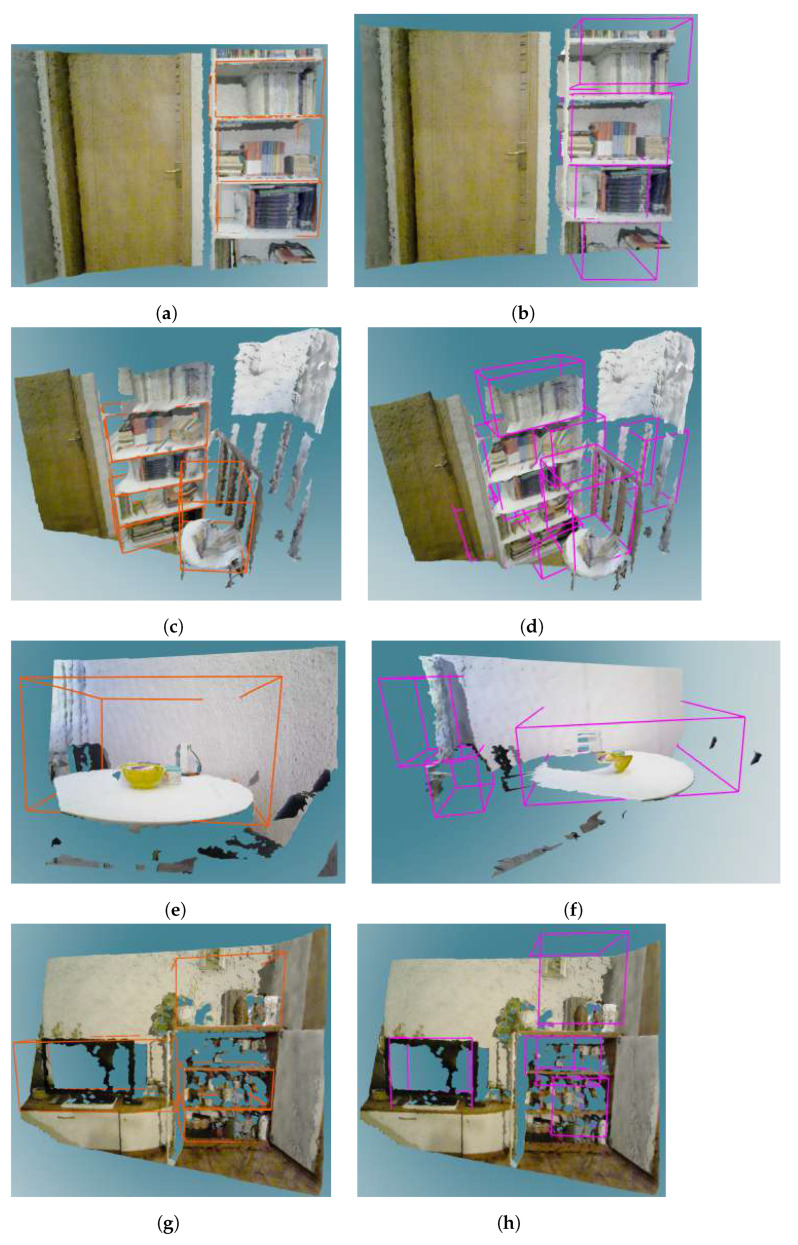
Examples of storage volumes in depth images contained in a real dataset. The left column shows ground truth annotations of storage volumes in the selected furniture classes. The right column shows the storage volumes predicted in these furniture models by the VoteNet detector. (**a**,**b**) Show a close-up view of a library bookshelf, (**c**,**d**) Show a library bookshelf seen from a distance, (**e**,**f**) Show a dining table, (**g**,**h**) Show a scene from the Family room.

**Table 1 sensors-22-06774-t001:** Distribution of the generated synthetic dataset per furniture class.

		Count of Generated Scenes				
Class Name	Count of Selected *Shapenet* Models	Training	Validation	Evaluation	Total	Count of Labeled Storage Volumes
bookshelf	41	3600	901	501	5002	26,841
cabinet	10	725	182	501	1008	2702
desk	18	725	182	101	1008	2142
table	34	712	179	99	990	1512
table empty	34	712	179	99	990	1530
negative	4	720	180	100	1000	0
**TOTAL**	**141**	**7194**	**1803**	**1001**	**9998**	**34,727**

**Table 2 sensors-22-06774-t002:** Minimum, maximum, and median volume of annotated storage space with corresponding size in each direction for each furniture class in the synthetic dataset. Column “median per axis” shows the median length, height, and width of each furniture class.

	Bounding Box														
Class Name	Minimum				Maximum				Median All				Median per Axis		
	Size [m]			Volume	Size [m]			Volume	Size [m]			Volume	Size [m]		
	*x*	*y*	*z*	[m${}^{3}$]	*x*	*y*	*z*	[m${}^{3}$]	*x*	*y*	*z*	[m${}^{3}$]	*x*	*y*	*z*
bookshelf	0.072	0.072	0.070	45 × 10^−6^	1.36	3.39	1.03	0.60	0.273	0.518	0.322	0.006	0.31	0.41	0.32
cabinet	0.074	0.096	0.078	69 × 10^−6^	1.30	4.86	1.98	1.56	0.273	0.268	0.220	0.002	0.21	0.33	0.23
desk	0.084	0.081	0.112	95 × 10^−6^	1.79	3.27	1.36	1.00	0.250	0.225	0.949	0.007	0.36	0.64	0.32
table	0.088	0.058	0.062	40 × 10^−6^	2.44	2.96	1.44	1.29	0.487	0.541	0.384	0.013	0.50	0.88	0.27
empty table	0.073	0.041	0.056	21 × 10^−6^	2.11	2.56	1.24	0.84	0.481	0.535	0.379	0.012	0.49	0.90	0.27

**Table 3 sensors-22-06774-t003:** Number of captured scenes and annotated storage volumes in the real dataset.

Environment	Home 1			Home 2				Office 1			Office 2	
Room/Furniture	Family Room	Living Room	Library	Bedroom	Gameroom	Kitchen	Pantry	Bookshelf Clutter	Bookshelf Empty	Desk	Cabinet	Total
**Count of captured scenes**	>4	5	12	23	8	5	22	16	11	36	9	151
**Count of annotated storage volumes**	17	11	43	54	20	5	88	51	55	102	34	480

**Table 4 sensors-22-06774-t004:** Minimum, maximum, and median storage volumes with corresponding size in each direction for each room in the real dataset. Column “median per axis” shows the median length, height, and width of each furniture class.

Real Dataset		Bounding Box														
Environment	Room	Minimum				Maximum				Median All				Median per Axis		
		Size [m]			Volume	Size [m]			Volume	Size [m]			Volume	Size [m]		
		*x*	*y*	*z*	[m${}^{3}$]	*x*	*y*	*z*	[m${}^{3}$]	*x*	*y*	*z*	[m${}^{3}$]	*x*	*y*	*z*
Home 1	Family room	0.374	0.368	0.232	0.004	1.18	0.65	0.52	0.05	0.673	0.381	0.279	0.009	0.38	0.40	0.31
	Kitchen	0.588	0.370	0.289	0.008	1.60	0.44	0.82	0.07	0.780	0.404	0.288	0.011	0.78	0.39	0.30
	Library	0.693	0.161	0.291	0.004	0.41	0.43	0.51	0.01	0.749	0.267	0.290	0.007	0.71	0.28	0.29
Home 2	Bedroom	0.550	0.051	0.277	0.001	1.506	0.471	0.553	0.049	0.516	0.390	0.271	0.007	0.534	0.394	0.280
	Gameroom	0.470	0.222	0.189	0.002	1.135	1.083	0.504	0.077	0.513	0.274	0.208	0.004	0.488	0.318	0.197
	Kitchen	0.892	1.051	0.500	0.059	1.190	1.017	0.629	0.095	1.109	1.052	0.500	0.073	1.082	1.052	0.500
	Pantry	0.257	0.078	0.128	0.000	1.169	0.330	0.271	0.013	0.730	0.267	0.263	0.006	0.740	0.271	0.259
Office 1	Bookshelf clutter	0.758	0.574	0.202	0.011	1.145	0.573	0.500	0.041	0.756	0.591	0.322	0.018	0.761	0.573	0.322
	Bookshelf empty	0.800	0.452	0.210	0.010	1.548	0.600	0.500	0.058	0.775	0.573	0.314	0.017	0.761	0.578	0.316
	Desk	0.240	0.666	0.087	0.002	1.328	0.670	0.535	0.060	0.218	0.764	0.644	0.013	0.614	0.637	0.502
Office 2	Cabinet	0.291	0.394	0.324	0.005	0.689	0.458	0.358	0.014	0.690	0.417	0.315	0.011	0.641	0.410	0.351

**Table 5 sensors-22-06774-t005:** Results of evaluation of the storage space detectors on the synthetic and real datasets. NMS IoU = 0.1; confidence threshold = 0.75. DS1 = bookshelves and negative scenes. DS2 = cabinets, desks, and (empty) tables. DS = complete dataset. Bold values represent higher precision or recall, which indicates a better result.

IoU Threshold	Metric	NN	Synthetic Dataset			Real Dataset
			DS1	DS2	DS	
0.25	Precision [%]	VoteNet	44.09	39.63	43.66	**42.22**
		FCAF3D	**45.47**	**57.69**	**47.72**	**24.17**
	Recall [%]	VoteNet	62.91	57.46	62.29	66.67
		FCAF3D	**70.83**	**73.31**	**71.41**	**75.42**
0.50	Precision [%]	VoteNet	13.85	13.99	14.03	5.25
		FCAF3D	**29.17**	**43.27**	**32.23**	**9.9**
	Recall [%]	VoteNet	35.77	35.78	35.91	21.04
		FCAF3D	**42.97**	**55.24**	**45.82**	**41.46**

**Table 6 sensors-22-06774-t006:** Results of evaluation of the storage volume detectors on the real dataset. Bold values represent higher precision or recall, which indicates a better result.

Environment			Home 1			Home 2				Office 1			Office 2	Dataset
Room		NN	Family Room	Living Room	Library	Bedroom	Gameroom	Kitchen	Pantry	Bookshelf Clutter	Bookshelf Empty	Desk	Cabinet	
**0.25**	Precision [%]	VoteNet	**37.73**	**56.44**	**43.22**	**33.64**	**40.72**	73.10	43.38	**64.50**	**53.21**	42.85	55.35	**42.22**
		FCAF3D	25.15	9.66	37.75	27.25	26.11	**74.58**	**45.72**	29.30	18.82	**53.41**	20.89	24.17
	Recall [%]	VoteNet	64.71	**81.82**	74.42	64.81	60.00	100.00	62.50	**84.31**	61.82	63.73	**91.18**	66.67
		FCAF3D	**82.35**	72.73	**76.74**	**85.19**	**65.00**	100.00	**78.41**	72.55	**65.45**	**72.55**	67.65	**75.42**
**0.50**	Precision [%]	VoteNet	0.84	0.00	0.83	4.61	0.00	**34.00**	0.73	**20.22**	1.88	6.32	**22.20**	5.25
		FCAF3D	**7.72**	**0.25**	**21.53**	**16.09**	**2.25**	27.50	**19.38**	17.66	**2.31**	**27.88**	8.27	**9.90**
	Recall [%]	VoteNet	5.88	0.00	11.63	20.37	0.00	60.00	6.82	39.22	9.09	24.51	**52.94**	21.04
		FCAF3D	**23.53**	**9.09**	**37.21**	**51.85**	**35.00**	**80.00**	**40.91**	**45.10**	**20.00**	**49.02**	41.18	**41.46**

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
