# Peer review of "Detection of Household Furniture Storage Space in Depth Images"

_sensors, 2022, doi:10.3390/s22186774_

Round 1

Reviewer 1 Report

Title: Detection of household furniture storage space in depth images

Review:

The author aims to detect the household furniture storage space in depth images, the topic is interesting although some problems need to be addressed well.

I have some major and minor questions:

For the major questions:

1.       The detection regions are considered cuboids, which is relevant for synthetic data where the image is well captured. But in real life, this is hard to get. The conditions are more complicated where the cupboard is half full, or already put some things inside it. How does the author determine whether the storage is available or not in cuboids term?

2.       The model is built based on ShapeNet and VoteNet, I don’t see an improvement in the methods. The authors only use the existing methods to detect based on the domain application. I don’t think it is enough to enhance the contributions in the neural network, especially object detection and artificial intelligence.

3.       I think the author needs to draw a system architecture to present a better presentation of the experiments conducted by the author. The author also doesn’t state the neural network VoteNet parameter clearly. Is the author using the same hyperparameter setting with the original VoteNet? Like I said before, the author merely uses an existing model in a new domain model.

4.       To detect the storage spaces the model needs to obtain two images, one is the front view, and the second is the side view. Correct me if I am wrong, but this kind of method needs to process more computation,  but the author doesn’t state the urgency of this problem.

5.       The author needs to state clearly the step-by-step detection process, like when to generate depth images, and what is depth image. How depth images are generated and so on. Many unclear steps appear in the manuscript.

Minor questions:

1.       The author needs to give more relevant references, and better no more than 3 years.

2.       The author needs to ref. the equations which are used in the manuscript.

3.       What are point clouds? No explanation for it. Suddenly appears in Figure 10.

4.       The presentation of the paper needs to be modified so that the reader easy to follow.

Reviewer 2 Report

This paper presents a neural network-based computer vision method for the detection of storage space within storage furniture. The method consists of automatic storage volume detection and annotation within 3D models of furniture and automatic generation of a large number of depth images of storage furniture with assigned bounding boxes representing the storage space above the furniture shelves.

-In academic work, comparing the obtained results to some related/recently published works under the same conditions (i.e., databases + protocols of evaluation) is necessary. The objective is to show the superiority of the presented work against the existing ones. Please explain more about the previous research result in this field.

-It is suggested to move Figure 1. The method overview in methodology.

- Please revised Figure 12, separate the graph will be better. It is also suggested to remove some Figures in the manuscript and show only the important Figure.

- Actually, the result in this paper is still low. The author state "Overall, the trained network achieves storage volume detection precision of 43.66% and recall 62.29% on a synthetic dataset with IoU threshold of 0.25." The author needs to do the experiment again and improve the experiment result.

-Which neural network do you use to train the model? Yolo 4, Yolo 5, RNN, Faster CNN? Explain in detail.

-What is the format of your dataset? How do you label your dataset?

-The author state that one of their contributions is "A dataset containing synthetic and real depth images with annotated ground truth storage volume bounding boxes." The discussion about this dataset is not clear enough. For example, what format do you use? XML, txt, JSON?

- Add more related references.

Reviewer 3 Report

You have done a really good job. I would only add some more engineering and specifically computer science contributions by adding more algorithms like Algorithm 1. It/they would show what and how exactly the other steps and stages of the whole procedure take place in the procedure of implementation and the adjacent experiments, beyond the space detection. This would indeed enhance your work and make it more interesting to computer scientists and engineers.

Round 2

Reviewer 2 Report

Thank you author for revising this paper. I still have some suggestions as follows:

- It is suggested to move Figure 1. The method overview in methodology.

- Please revised Figure 12, separate the graph will be better or give a, b, c, d..etc. It is also suggested to remove some Figures in the manuscript and show only the important Figure.

- Please explain the VoteNet and FACAF3D neural networks in detail, did you do any modification? 
